# Subgroup evaluation to understand performance gaps in deep learning-based classification of regions of interest on mammography

**MinJae Woo** [1]ᴼ*, **Linglin Zhang** [2]ᴼ‡, **Beatrice Brown-Mulry** [3], **InChan Hwang**[4], **Judy Wawira Gichoya**[5], **Aimilia Gastounioti** [6], **Imon Banerjee**[7,8], **Laleh Seyyed-Kalantari**[9], **Hari Trivedi**[5]

**1** Department of Public Health Sciences, Clemson University, Clemson, South Carolina, United States of America, **2** Artificial Intelligence Ethics Laboratory, Equifax Inc., Alpharetta, Georgia, United States of America, **3** Department of Computer Science, Emory University, Atlanta, Georgia, United States of America, **4** Department of Computer Science, Montreat College, Montreat, North Carolina, United States of America, **5** Department of Radiology and Imaging Sciences, Emory University, Atlanta, Georgia, United States of America, **6** Computational Imaging Research Center, Washington University in St. Louis School of Medicine, St. Louis, Missouri, United States of America, **7** Department of Radiology, Mayo Clinic Arizona, Scottsdale, Arizona, United States of America, **8** School of Computing and Augmented Intelligence, Arizona State University, Tempe, Arizona, United States of America, **9** Department of Electrical Engineering and Computer Science, York University, Toronto, Ontario, Canada

ᴼ These authors contributed equally to this work.
‡ This author is co-first author on this work.
* mwoo@clemson.edu

## Abstract

This study evaluates a deep learning model for classifying normal versus potentially abnormal regions of interest (ROIs) on mammography, aiming to identify imaging, pathologic, and demographic characteristics that may induce suboptimal model performance in certain patient subgroups. We utilized the EMory BrEast imaging Dataset (EMBED), containing 3.4 million mammographic images from 115,931 patients. Full-field digital mammograms from women aged 18 years or older were used to create positive and negative patches with the patches matched based on size, location, patient demographics, and imaging features. Several convolutional neural network (CNN) architectures were tested, with ResNet152V2 demonstrating the best performance. The dataset was split into training (29,144 patches), validation (9,910 patches), and testing (13,390 patches) sets. Performance metrics included accuracy, AUC, recall, precision, F1 score, false negative rate, and false positive rate. Subgroup analysis was conducted using univariate and multivariate regression models to control for confounding effects. The classification model achieved an AUC of 0.975 and a recall of 0.927. False negative predictions were significantly associated with White patients (RR = 1.208; p = 0.050), those never biopsied (RR = 1.079; p = 0.011), and cases with architectural distortion (RR = 1.037; p < 0.001). Higher breast density significantly increased the risk of false positives, with BI-RADS density C (RR = 1.891; p < 0.001) and D (RR = 2.486; p < 0.001). Race and age were not significant predictors for false positives in multivariate analysis. These findings suggest that deep learning models

**Data availability statement:** The dataset used in this study contains real patient imaging data and is subject to ethical and legal restrictions to protect patient privacy. A portion of the dataset is publicly available through the AWS Open Data Registry, where researchers can request access following the established review process: https://registry.opendata.aws/emory-breast-imaging-dataset-embed/. For access to additional data beyond the publicly available portion, researchers should submit their request through the same form provided for the public dataset, clearly indicating the need for full data access. A member of the data management team will follow up to guide the requester through the official data access procedures, which are governed by institutional policies and data use agreements.

**Funding:** This study was funded by the Equifax Ethical AI Grant (A23-0024) and the National Institutes of Health R01 Grant (R01CA286120). The funders had no role in the study design, data collection, analysis, decision to publish, or preparation of the manuscript.

**Competing interests:** The authors have declared that no competing interests exist.

for mammography may underperform in specific subgroups. The study underscores the need for more precise patient subgroup analysis and emphasizes the importance of considering confounding factors in deep learning model evaluations. These insights can help develop fair and interpretable decision-making models in mammography, ultimately enhancing the performance and equity of CADe and CADx applications.

## Author summary

Our study reveals that AI models used in screening mammogram do not fail randomly, resulting in certain patient populations more prone to experiencing these failures. Specifically, deep learning models may underperform in specific patient subgroups such as those with higher breast density or certain demographic characteristics. To effectively address these failures, it is crucial to consider how patient characteristics (e.g., breast density, age, and race) are interrelated with one another. For example, while many researchers traditionally focus on augmenting datasets with images from minority racial groups to reduce model failures against them, our results indicate that increasing the representation of images from patients with higher breast density would be a more targeted and impactful approach after accounting for interaction between density and race. Ignoring the interconnected nature of these patient characteristics can lead to misguided interventions, possibly interfering with the efforts toward improving AI model effectiveness and fairness in clinical practice.

## Introduction

Breast cancer is the most common cancer among women, leading to approximately 42,000 deaths annually in the United States [1]. Early detection via screening mammography has been shown to reduce morbidity and mortality of breast cancers by 38–48% [2–4]. However, there is significant resource usage for detecting a relatively small number of cancers; for every 1,000 women screened, approximately 100 are recalled, and 30–40 are biopsied for the detection of up to 1–5 cancers [5]. This is largely due to the similar imaging appearance of many benign and malignant lesions which necessitates repeat imaging and/or biopsy for definitive diagnosis. The challenge of distinguishing normal from suspected abnormal findings on screening mammography can lead to unnecessary recalls and biopsies. Improving the differentiation of these findings has the potential to reduce resource wastage and decrease patient morbidity [3,6,7].

Deep learning (DL) models are increasingly utilized to enhance the differentiation between normal and suspected abnormal regions of interest (ROIs) in mammograms [8–18]. Such models have strong potential for being directly implemented into computer-aided detection (CADe) and computer-aided diagnosis (CADx) [19–24]. Moreover, models that classify normal or suspected abnormal regions of interest often serve as the backbone of broader AI applications in cancer screening and diagnosis, such as tumor object detection and automated radiology report generation [9,10,15,25,26]. However, these models are imperfect, generally achieving sensitivity and specificity between 85–92%, and it has been well-documented that their failure does not occur at random [27–30]. Because breast cancer can have a heterogenous appearance [31], we hypothesize that various image features or other image confounders result in asymmetric model performance for subgroups of lesion appearance, pathology, or

patient demographics, and that these confounders may be interrelated. The collective influence of multiple imaging features, breast densities, and patient demographics on the model failure remains largely unexamined due to the lack of available granular datasets in the field. This creates potential unrecognized gaps in DL model performance and failure, which may disproportionately affect specific patient subgroups and undermine the overall effectiveness as well as equity of diagnostic tools in the era of artificial intelligence.

In this paper, we describe a deep learning model for classifying normal versus potentially abnormal regions of interest on mammography, with purpose to present a robust, multivariate post-hoc analysis to identify imaging, pathologic, and demographic characteristics that degrade model performance. The presented analysis aims to determine which patient subgroups truly suffer from disparate impacts due to model failures. To this end, we leveraged the largest mammography dataset in the field, which is racially balanced and of high granularity and quality.

## Materials and methods

### Data preparation

We used the EMory BrEast imaging Dataset (EMBED), which contains clinical and imaging data for 3.4 million mammographic images from 383,379 screening and diagnostic exams of 115,931 patients, acquired between 2013 and 2020 at four different hospitals within a single academic institution [32]. With the approval of Emory University's institutional review board, this dataset was curated retrospectively using automated and semi-automated curation techniques, and the need for written informed consent from patients was waived due to the use of de-identified data. The dataset provides comprehensive detailed demographics, imaging features, pathologic outcomes, and Breast Imaging Reporting and Data System (BI-RADS) tissue densities and scores for both screening and diagnostic exams. Additionally, it provides regions of interest (ROIs) annotated by radiologists during initial screenings, along with more definitive diagnostic information from follow-up diagnostic or pathologic studies.

An overview of the data preparation process is shown in Fig 1. This study exclusively utilized full-field digital mammograms (FFDM) from women aged 18 years or older with at least one available mammogram in the picture archiving and communication system. Positive patches were defined as any radiologist-annotated region of interest (ROI) from a Breast Imaging Reporting and Data System (BI-RADS) 0 image, indicating a potential abnormality requiring follow-up. Negative patches were generated from two sources: (1) regions within the BI-RADS 0 images that did not overlap with the annotated ROIs; and (2) regions from negative screening images classified as BI-RADS 1 and 2, as illustrated in Fig 2. We employed a stratified down-sampling approach to minimize confounding factors in patch selection. Given that the pool of patients available for negative patch extraction is much larger than that for positive patches, we carefully subsampled negative patches to ensure that the patient distribution used for their extraction closely matched that of positive patches. Specifically, this process was to control for key confounders, including patient demographics (e.g., race), imaging features, and breast density, preventing the model from exploiting unintended differences in patient composition rather than learning clinically relevant imaging patterns. The subsequent matching process first identified pairs of positive and negative FFDMs with the highest similarity in terms of pixel distributions, then used the shape and location of the ROI from the positive images as references, and finally extracted equivalent patches from the negative images. The extracted patch sizes varied from $53 \times 76 - 2379 \times 2940$ pixels, with a median size of $360 \times 437$ pixels. Patches larger than $512 \times 512$ pixels were downsampled to fit within $512 \times 512$ pixels while preserving the aspect ratio. All patches were then zero-padded to a consistent

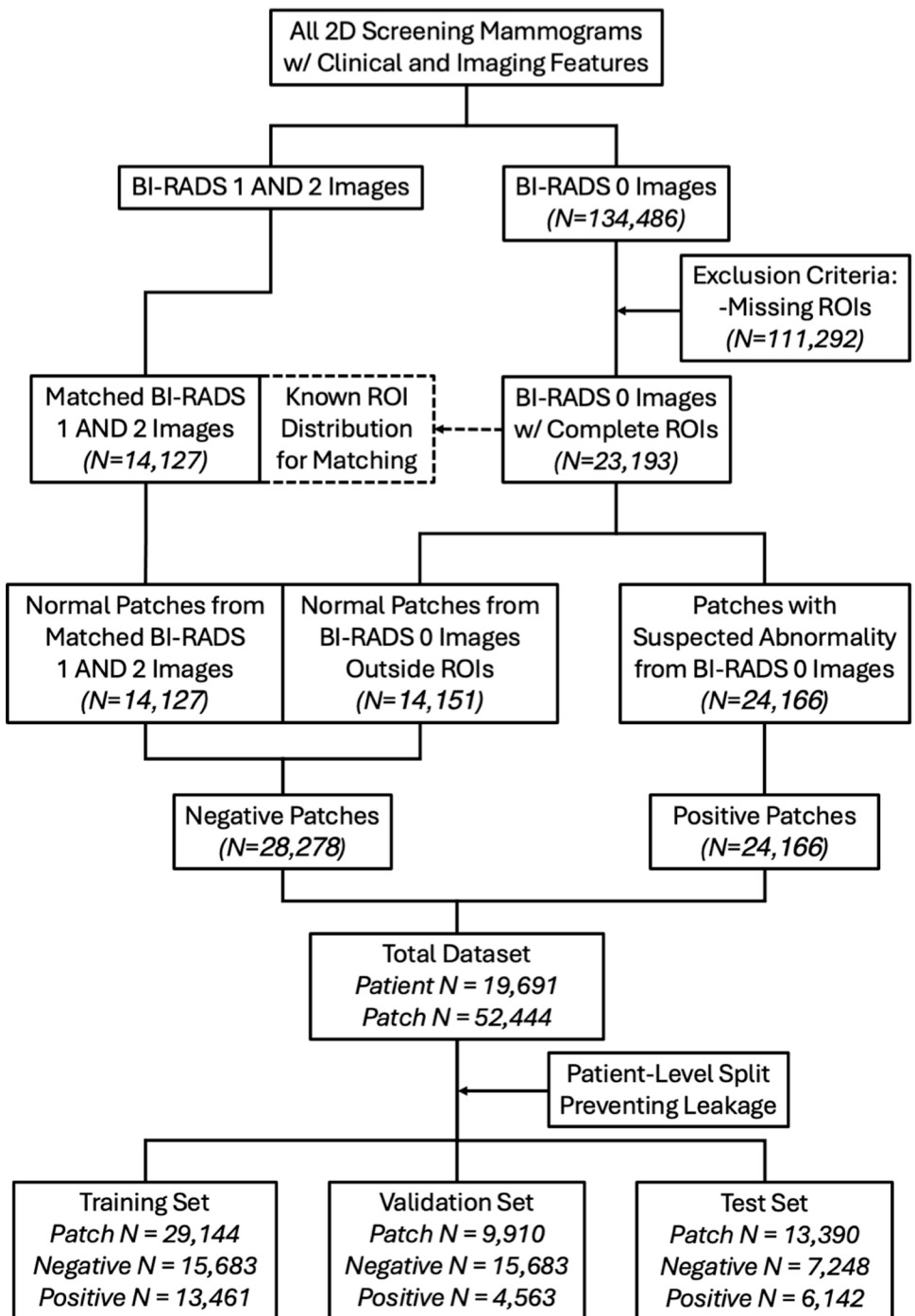

**Fig 1. Overview of data preparation for DL patch classification model training.** Positive patches were radiologist-annotated ROIs from BI-RADS 0 images, while negative patches were acquired from non-overlapping regions in BI-RADS 0 images and BIRADS 1 and 2 images. Negative patches were matched to positive ones by size, location, and patient demographics. DL, deep learning; BI-RADS, breast imaging reporting and data system.

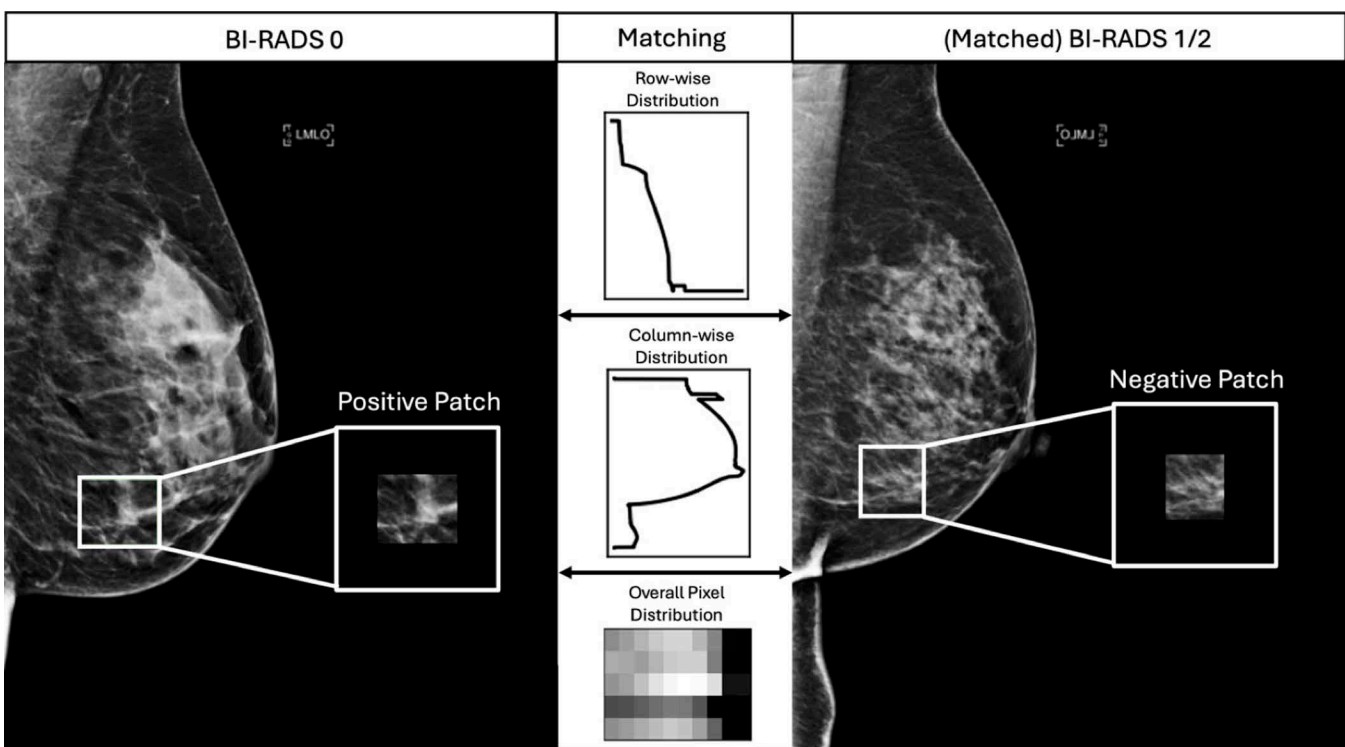

**Fig 2. Example of positive and negative patch generation.** The patch in the BI-RADS 0 image (left) shows a suspected abnormality originally annotated by the radiologist. The patches in the matched BI-RADS 1/2 image (right) represent an equivalent patch which serves as a negative counterpart. The matching process utilized row-wise and column-wise distribution along with overall pixel distribution to maximize similarity in the overall intensity distributions between negative and positive patches. All patches were cropped, centered, and padded with 0 pixels to be 512 × 512 pixels. BI-RADS, breast imaging reporting and data system.

input size of 512 × 512 pixels for compatibility with the convolutional neural network (CNN) architectures. The final dataset was randomly split at the patient level to prevent data leakage, comprising a total of 10,678 patients and 29,144 patches (55.6%) for training, 3,609 patients and 9,910 patches (18.9%) for validation, and 5,404 patients and 13,390 patches (25.5%) for testing.

## Patch classifier model training

Guided by relevant studies, we tested four common CNN architectures and their corresponding pretrained weights as feature extractors in the binary classification of normal and potentially abnormal tissue patches: InceptionV3 [33], VGG16 [34], ResNet50V2 [35], and ResNet152V2 [36], using identical training, validation, and test sets. Preliminary results demonstrated the best performance demonstrated by ResNet152V2 (S1 Table), so this architecture was utilized throughout the study. The final architecture was equipped with input layers adjusted to accept 512 × 512-pixel resolution PNG format images, followed by the ResNet152V2 feature extractor pretrained on the ImageNet dataset [37], and additional layers including a trainable batch normalization layer; the following deep neural network constituents included a dense layer with ReLU activation, a dropout layer, three dense layers with ReLU, and final dense layer activated by Sigmoid function. The systematic optimization of hyperparameters and learning rate was performed using Bayesian optimization [38], resulting in optimal model validation performance at a learning rate of 0.0059. The hardware

configuration used for the experiment was (1) 8 x NVIDIA A100 GPUs providing a combined total of 320 GB of GPU RAM, and (2) 3 x NVIDIA RTX3090 GPUs providing a combined total of 72 GB of GPU RAM, utilized as appropriate. All data preprocessing and deep learning model development were conducted in a Python environment using TensorFlow. Statistical analyses were performed using the statsmodels package in Python and cross-validated using StataIC 15.1 (StataCorp, College Station, TX, USA).

## Statistical analysis

We evaluated overall patch classifier performance by accuracy, area under the receiver operating characteristics curve (AUC), recall, precision, F1 score, false negative rate, and false positive rate. To obtain 95% confidence intervals, we performed bootstrapping with two hundred iterations of the test set, guided by relevant studies [39–41].

The experiment consisted of two steps: (1) the typical process of developing and testing a deep learning model for the binary classification of normal and potentially abnormal tissue patches, and (2) analysis to identify patient subgroups more likely to be misclassified by the model, Fig 3. For subgroup evaluation, we calculated AUC, recall, and precision by race – White, Black, or Other; age groups – < 50, 50–60, 60–70, and >70 years; BI-RADS tissue density – A, B, C, or D; pathology – never biopsied, cancer, or benign; and imaging features – mass, asymmetry, architectural distortion, or calcification. However, because multiple mammographic features are known to be interrelated (for example, breast density distribution varies by race and decreases with age), standard statistical evaluation of subgroups using a student's t-test would yield spurious results as this test assumes independence of all variables. Instead, we used univariate and multivariate logistic regression models to evaluate the impact of various demographic, imaging, and pathologic subgroups on model performance. This allowed us to control for potential confounding effects between features and could help to explain the underlying contribution of each feature to model failure. False negative predictions were investigated on the potentially abnormal patches using race, age, tissue density, pathological outcome, and

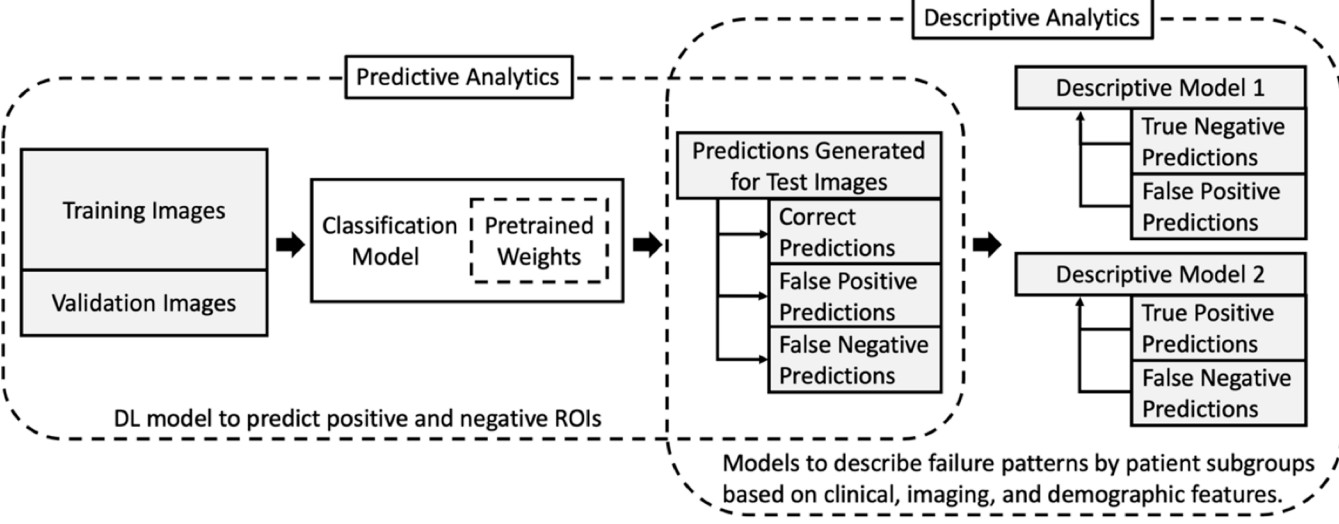

**Fig 3. Experimental overview with two main components.** Predictive Analytics involved training a deep learning model to differentiate between normal and potentially abnormal patches on mammography. Descriptive Analytics evaluated model performance by subgroups based on clinical, imaging, and demographic features. Descriptive Model 1 analyzed false positives, and Descriptive Model 2 analyzed false negatives. DL, deep learning; ROI, region of interest.

image findings. False positive predictions were evaluated on normal patches only using features that would apply to the whole mammogram - race, age group, and tissue density. The odds ratio (OR) for false positive and false negative predictions was calculated between each feature and a selected control group model, and the subsequent risk ratio (RR) was calculated by OR and the non-exposed (correct prediction result; positives within all positive patches and true negative within all negative patches) prevalence of the current feature.

## Results

### Patient characteristics

Mean patient age was 59.02 ± 11.87 (standard deviation) years. The study included 21,273 (40.6%) White patients, 22,321 (42.5%) Black patients, and 8,850 (16.9%) patients from other racial backgrounds. 16,387 (31.2%) patients were under 50 years old, 14,775 (28.2%) were between 50–60 years old, 12,490 (23.8%) were between 60–70 years old, and 8,792 (16.8%) were over 70 years old. The tissue density distribution for BI-RADS categories A, B, C, and D was 6,050 (11.5%), 18,544 (35.4%), 23,218 (44.3%), and 4,632 (8.8%) respectively. The distribution of demographics, imaging features, and pathologic outcomes are summarized in Table 1. Pathologic outcomes stratified by type of imaging finding in the test set are shown in Table 2.

**Table 1. Distribution of regions of interest (patches) utilized for model development.**

| Characteristics | Count (percentage) | | | |
|---|---|---|---|---|
| | Total | Training set | Validation set | Test set |
| All patches | 52,444 (100%) | 29,144 (100%) | 9,910 (100%) | 13,390 (100%) |
| **Label** | | | | |
| Positive | 24,166 (46.1%) | 13,461 (46.2%) | 4,563 (46.0%) | 6,142 (45.9%) |
| Negative | 28,278 (53.9%) | 15,683 (53.8%) | 5,347 (54.0%) | 7,248 (54.1%) |
| **Race** | | | | |
| African American or Black | 22,321 (42.5%) | 12,483 (42.8%) | 4,137 (41.7%) | 5,701 (42.6%) |
| Caucasian or White | 21,273 (40.6%) | 11,649 (40.0%) | 4,140 (41.8%) | 5,484 (40.9%) |
| Other | 8,850 (16.9%) | 5,012 (17.2%) | 1,633 (16.5%) | 2,205 (16.5%) |
| **Age group** | | | | |
| < 50 | 16,387 (31.2%) | 9,184 (31.5%) | 3,293 (33.2%) | 3,910 (29.2%) |
| 50–60 | 14,775 (28.2%) | 8,138 (27.9%) | 2,811 (28.4%) | 3,826 (28.6%) |
| 60–70 | 12,490 (23.8%) | 6,965 (23.9%) | 2,240 (22.6%) | 3,285 (24.5%) |
| >70 | 8,792 (16.8%) | 4,857 (16.7%) | 1,566 (15.8%) | 2,369 (17.7%) |
| **BI-RADS density** | | | | |
| A-Almost entirely fatty | 6,050 (11.5%) | 3,218 (11.0%) | 1,166 (11.8%) | 1,666 (12.4%) |
| B-Scattered areas of fibro-glandular density | 18,544 (35.4%) | 10,275 (35.3%) | 3,348 (33.8%) | 4,921 (36.8%) |
| C-Heterogeneously dense | 23,218 (44.3%) | 12,682 (43.5%) | 4,310 (43.5%) | 6,226 (46.5%) |
| D-Extremely dense | 4,632 (8.8%) | 2,969 (10.2%) | 1,086 (10.9%) | 577 (4.3%) |
| **Pathology** | | | | |
| Cancer | 1,147 (2.2%) | 739 (2.5%) | 233 (2.3%) | 175 (1.3%) |
| Benign | 3,495 (6.7%) | 2,025 (7.0%) | 662 (6.7%) | 808 (6.0%) |
| Never biopsied | 47,802 (91.1%) | 26,380 (90.5%) | 9,015 (91.0%) | 12,407 (92.7%) |

Due to the structure of the dataset, the prevalence of imaging features could not be calculated for the training and validation datasets; they may contain multiple ROIs, which prevents the calculation of the exact prevalence.

BI-RADS, breast imaging reporting and data system.

**Table 2. Distribution of pathologic outcomes stratified by image findings in the test set.**

| Image findings | Cancer | Benign | Never biopsied | Total |
|---|---|---|---|---|
| Mass | 8 (0.45%) | 75 (4.22%) | 1,693 (95.33%) | 1,776 (100%) |
| Asymmetry | 97 (3.86%) | 390 (15.54%) | 2,023 (80.60%) | 2,510 (100%) |
| AD | 19 (3.28%) | 50 (8.64%) | 510 (88.08%) | 579 (100%) |
| Calcification | 82 (1.53%) | 265 (4.93%) | 5,025 (93.54%) | 5,372 (100%) |

AD, architectural distortion.

## Patch classification performance analysis

On the test set of 13,390 patches from 5,404 patients, the patch classification model built with ResNet152V2 achieved an overall accuracy of 92.6% (95% CI: 92.0–93.2%), an AUC of 0.975 (95% CI: 0.972–0.978), a recall of 0.927 (95% CI: 0.919–0.935), a precision of 0.912 (95% CI: 0.902–0.922), and an F1 score of 0.919 (95% CI: 0.913–0.925) for classifying normal versus suspected abnormal patches, Table 3. A significant impact of breast density on classification

**Table 3. Classification model performance both overall and by subgroup.**

| Group | Accuracy | AUC | Recall | Precision | F1 score |
|---|---|---|---|---|---|
| **Overall** | | | | | |
| Train | 0.984 | 0.998 | 0.984 | 0.983 | 0.983 |
| Validation | 0.926 | 0.967 | 0.915 | 0.925 | 0.920 |
| Test | 0.926 ± 0.006 | 0.975 ± 0.003 | 0.927 ± 0.008 | 0.912 ± 0.010 | 0.919 ± 0.006 |
| **Race** | | | | | |
| White | 0.922 ± 0.009 | 0.972 ± 0.005 | 0.918 ± 0.013 | 0.902 ± 0.016 | 0.910 ± 0.011 |
| Black | 0.927 ± 0.009 | 0.976 ± 0.005 | 0.931 ± 0.012 | 0.914 ± 0.015 | 0.922 ± 0.010 |
| Other | 0.928 ± 0.015 | 0.978 ± 0.007 | 0.938 ± 0.020 | 0.926 ± 0.020 | 0.932 ± 0.014 |
| **Age** | | | | | |
| < 50 | 0.916 ± 0.012 | 0.970 ± 0.007 | 0.922 ± 0.014 | 0.919 ± 0.016 | 0.920 ± 0.011 |
| 50–60 | 0.926 ± 0.011 | 0.977 ± 0.005 | 0.930 ± 0.017 | 0.915 ± 0.018 | 0.922 ± 0.012 |
| 60–70 | 0.933 ± 0.013 | 0.976 ± 0.006 | 0.932 ± 0.017 | 0.916 ± 0.022 | 0.924 ± 0.016 |
| >70 | 0.930 ± 0.014 | 0.975 ± 0.008 | 0.928 ± 0.023 | 0.882 ± 0.031 | 0.904 ± 0.021 |
| **BI-RADS density** | | | | | |
| A | 0.950 ± 0.016 | 0.977 ± 0.013 | 0.924 ± 0.040 | 0.825 ± 0.069 | 0.871 ± 0.044 |
| B | 0.942 ± 0.009 | 0.982 ± 0.004 | 0.942 ± 0.012 | 0.936 ± 0.014 | 0.939 ± 0.009 |
| C | 0.910 ± 0.008 | 0.966 ± 0.005 | 0.920 ± 0.012 | 0.906 ± 0.013 | 0.913 ± 0.009 |
| D | 0.877 ± 0.036 | 0.953 ± 0.021 | 0.899 ± 0.046 | 0.882 ± 0.047 | 0.891 ± 0.034 |
| **Pathology** | | | | | |
| Never biopsied | 0.925 ± 0.006 | 0.974 ± 0.003 | 0.925 ± 0.008 | 0.908 ± 0.010 | 0.916 ± 0.007 |
| Benign | 0.937 ± 0.019 | 0.977 ± 0.010 | 0.955 ± 0.022 | 0.943 ± 0.027 | 0.949 ± 0.016 |
| Cancer | 0.930 ± 0.049 | 0.980 ± 0.023 | 0.938 ± 0.063 | 0.957 ± 0.052 | 0.947 ± 0.042 |
| **Image findings** | | | | | |
| Mass | 0.931 ± 0.017 | 0.980 ± 0.008 | 0.949 ± 0.022 | 0.896 ± 0.029 | 0.922 ± 0.020 |
| Asymmetry | 0.930 ± 0.009 | 0.977 ± 0.005 | 0.937 ± 0.010 | 0.942 ± 0.011 | 0.939 ± 0.008 |
| AD | 0.826 ± 0.032 | 0.914 ± 0.034 | 0.810 ± 0.040 | 0.939 ± 0.027 | 0.869 ± 0.026 |
| Calcification | 0.920 ± 0.016 | 0.974 ± 0.008 | 0.939 ± 0.018 | 0.904 ± 0.022 | 0.921 ± 0.016 |

The overall and subgroup AUC, recall, and precision averaged over 200 bootstrapped samples ± 95% confidence intervals.

AD, architectural distortion; AUC, area under the receiver operative characteristics curve; BI-RADS, breast imaging reporting and data system.

performance was observed; detailed classification model performance stratified by breast density across subgroups of race, age, tissue density, pathology outcome, and image findings are described in S2 Table. The typical subgroup analysis in the field, represented by ROC curves by subgroups, is shown in Fig 4. Another predominant subgroup analysis, represented by histograms showing distribution of AUCs with bootstraps showing the separation between groups, is shown in Fig 5. Some patches were manually inspected to ensure the rigor of the failure analysis; examples of True Positive, False Negative, True Negative, and False Positive prediction outcomes by various subgroups are shown in Fig 6.

The relative risk ratio of false negative (Table 4) and false positive (Table 5) predictions was calculated using univariate and multivariate logistic regression models. In sum, patients who faced false negatives were significantly more likely to be White (RR = 1.208; p = 0.050), never biopsied (RR = 1.079; p = 0.011), with no masses (RR = 1.086; p = 0.010), with no asymmetries (RR = 1.171; p = 0.040), and with architectural distortion (RR = 1.037; p < 0.001). Similarly, patients who faced false positives were significantly more likely to have higher breast density, specifically with density C (RR = 1.891; p < 0.001) and density D (RR = 2.486; p < 0.001). Patients who faced false negatives were significantly more likely to be White compared to other racial groups, but this association was not significant when specifically compared to Black patients. It is also noteworthy that race and age were no longer significant predictors for false positives when using multivariate analysis. This change in statistical significance was observed in both false negative and false positive analyses before and after addressing the potential confounding effects between demographic variables and image features. Generally, addressing confounding effects results in fewer significant predictors for both false positives and false negatives.

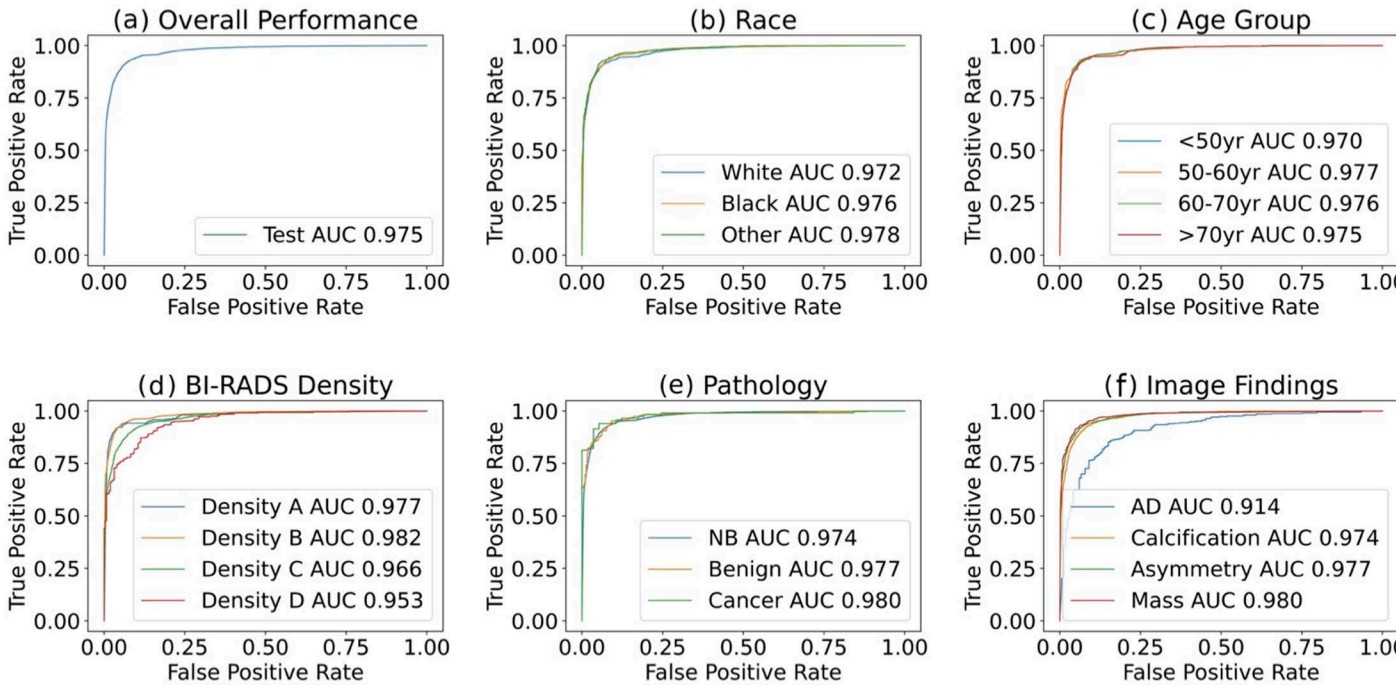

**Fig 4. Receiver operating characteristic curves of (A) the whole test set, and stratified by subgroups: (B) race, (C) age, (D) BI-RADS density, (E) pathology, and (F) image findings.** AUC, area under the receiver operative characteristics curve; BI-RADS, breast imaging reporting and data system; NB, never biopsied; AD, architectural distortion.

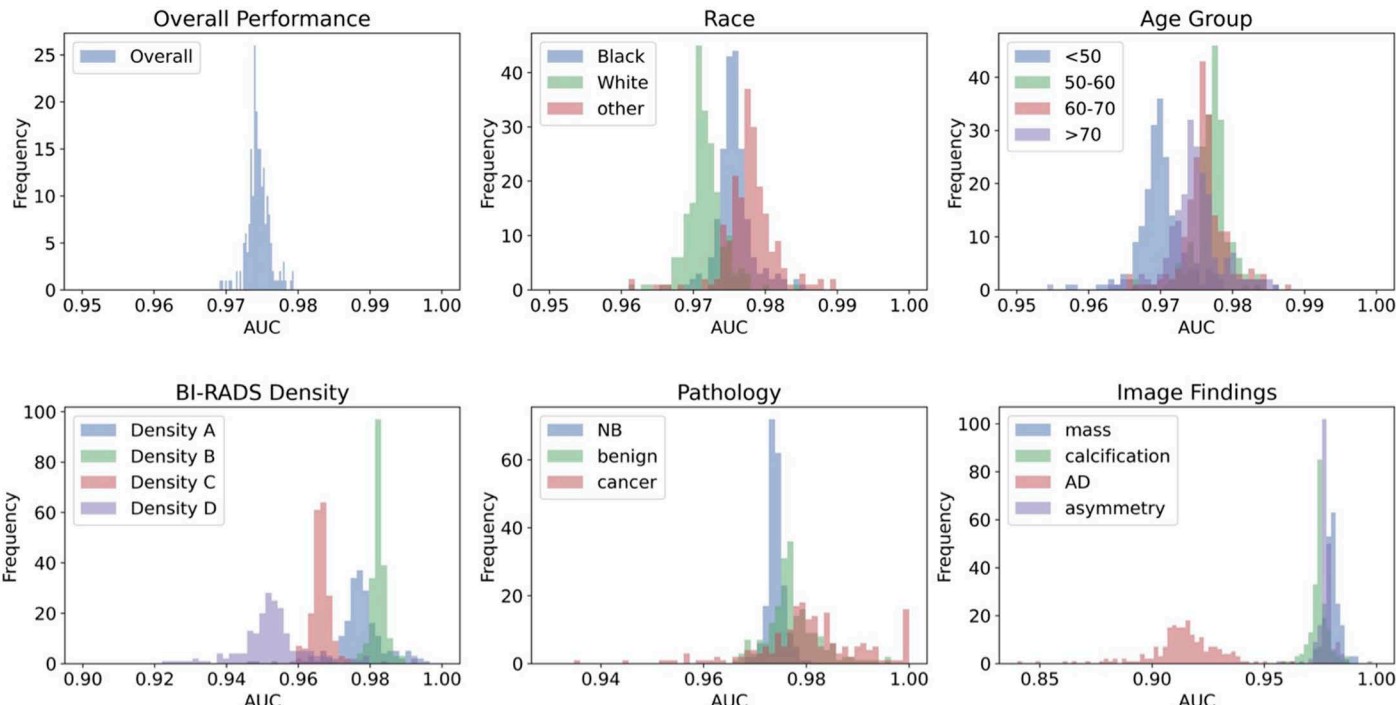

**Fig 5. Histogram demonstrating distribution of AUCs of the test set with 200 bootstraps showing the separation between subgroups within each category.**
AD, architectural distortion; AUC, area under the receiver operating characteristics curve; BI-RADS, breast imaging reporting and data system; NB, never biopsied.

## Discussion

Our study addresses several key factors in the use of deep learning models in screening mammography. First – that deep convolutional network architectures for abnormality classification underperform in certain subgroups of patients or imaging findings. This discrepancy is also likely to be present in other broader applications such as tumor object detection and automated radiology report generation, as they rely on DL models for ROI classification as key components in their backbone architecture. Second – that the mechanisms driving model underperformance in specific subgroups of demographics and imaging features are more complex than previously understood. Our evaluation provides the first subgroup evaluation of abnormality classification at the patch level while also considering confounding between groups. Most subgroup evaluation considers classes like race, age, imaging features, or pathologic outcomes as independent variables [42–44], when in reality, they are interrelated [28–30]. These results lend insight into where these models truly fail during clinical deployment. For example, we found that architectural distortion conferred a higher risk of false negatives, even after considering confounding with breast density, race, and age. We also learned that neither tissue density nor race exhibits a significant association with model failure defined by false negatives, but that breast density alone does significantly affect false positives; BI-RADS density C and D, have a 1.89 and a 2.48 times higher increased risk of false positives compared to BI-RADS density A patches, respectively. Overall, these results are consistent with our initial hypothesis suggesting that patterns in model failure are (1) different between types of failure, (2) subject to potential confounding effects. Recent studies indicate that even modest misclassifications by deep learning models can disrupt radiologists' diagnostic workflows and influence patient management decisions [45,46]. This work was part of ongoing efforts to enhance

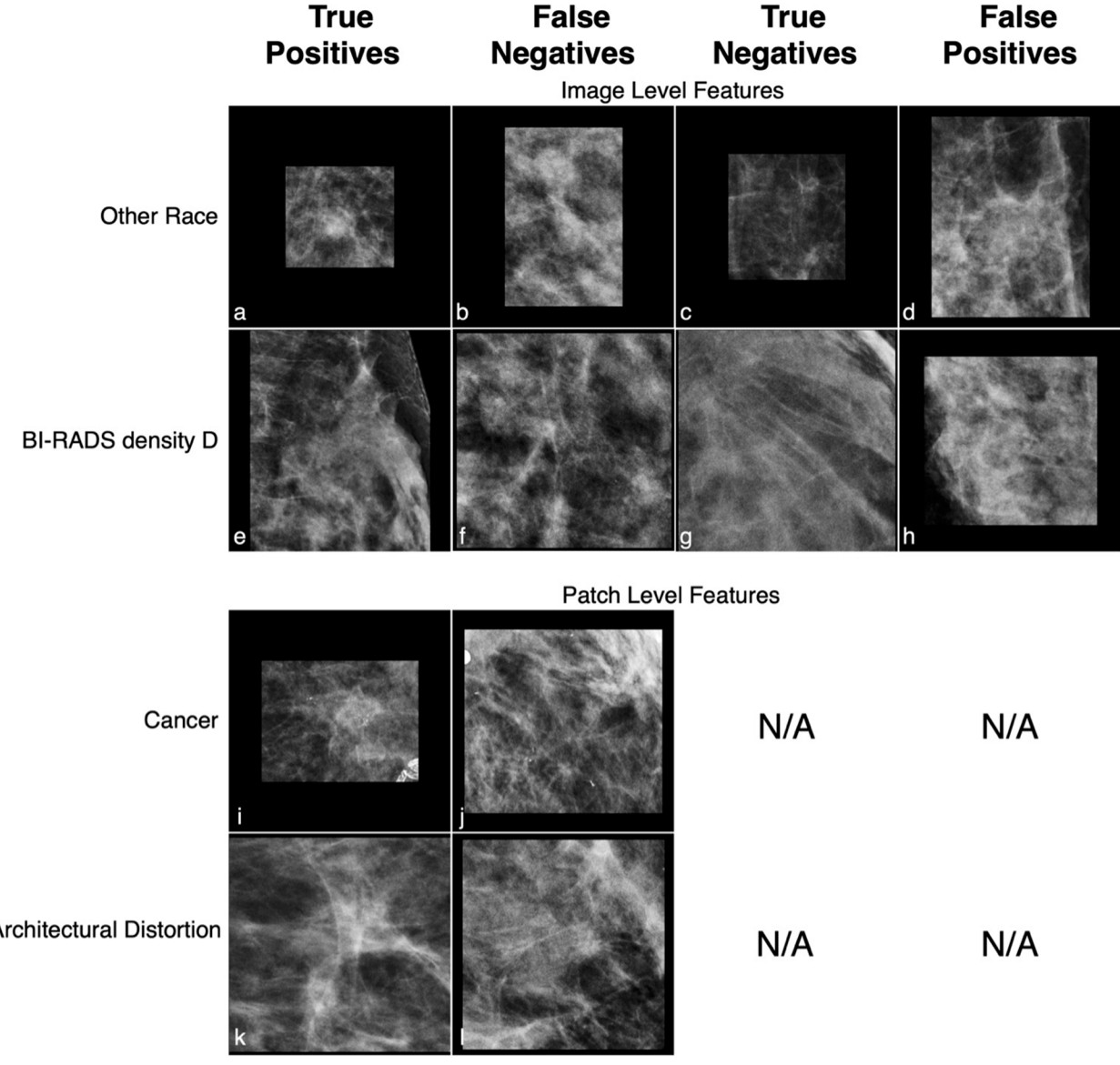

**Fig 6. Example of true positive, false negative, true negative, and false positive patches.** Each row is selected to demonstrate a single feature while holding other features constant. (a–d) are patches selected from patients of Other race, age less than 50 years old, BI-RADS density C, and never biopsied; (e–h) are patches of patients with density D, race White, age less than 50 years old, and never biopsied; (i and j) are patches with cancer, White, age greater than 70 years old, density B, with calcifications; (k and l) are patches with architectural distortion, White, age less than 50 years old, density C, never biopsied.

the clinical utility and fairness of deep learning-based mammography screening by identifying these error patterns in a rigorous way.

The findings should be interpreted considering the study's methodological constraints. First, this research was conducted at a single institution, utilizing medical images collected from multiple hospitals. Consequently, the extent to which our results can be generalized to other healthcare systems requires further investigation, which requires a dataset with similar annotated lesion characteristics at the patch level. Secondly, our study did not conduct a direct failure analysis on end-user applications. Instead, we hypothesize that the patient subgroup

**Table 4. Evaluation of model performance using multivariate regression to assess risk ratio of false negative predictions by subgroups, compared to versus univariate evaluation.**

| Variables | OR | RR | Univariate p-value | Multivariate p-value | Number of patches | Control group |
|---|---|---|---|---|---|---|
| Black | 0.880 [0.709, 1.093] | 0.922 [0.797, 1.056] | < 0.001* | 0.248 | 2,630 | White |
| Other* | 0.749 [0.561, 1.000] | 0.828 [0.673, 1.000] | <0.001* | 0.050* | 1,160 | White |
| 50–60 y/o | 0.881 [0.688, 1.127] | 0.918 [0.768, 1.081] | <0.001* | 0.315 | 1,798 | <50 y/o |
| 60–70 y/o | 0.823 [0.626, 1.082] | 0.875 [0.715, 1.053] | <0.001* | 0.163 | 1,434 | <50 y/o |
| >70 y/o | 0.89 [0.643, 1.231] | 0.924 [0.731, 1.143] | <0.001* | 0.482 | 840 | <50 y/o |
| Density B | 1.132 [0.421, 1.049] | 1.060 [0.434, 1.047] | <0.001* | 0.079 | 2,327 | Density A |
| Density C | 0.752 [0.564, 1.385] | 0.862 [0.577, 1.359] | 0.490 | 0.590 | 3,183 | Density A |
| Density D | 1.239 [0.618, 1.943] | 1.103 [0.630, 1.854] | 0.015* | 0.756 | 321 | Density A |
| Benign* | 0.567 [0.366, 0.881] | 0.927 [0.848, 0.986] | <0.001* | 0.011* | 499 | Never biopsied |
| Cancer | 0.778 [0.353, 1.714] | 0.971 [0.841, 1.045] | <0.001* | 0.533 | 118 | Never biopsied |
| Mass* | 0.596 [0.403, 0.882] | 0.921 [0.842, 0.983] | <0.001* | 0.010* | 761 | No mass |
| Asymmetry* | 0.751 [0.571, 0.988] | 0.854 [0.721, 0.994] | <0.001* | 0.040* | 3,127 | No asymmetry |
| AD* | 2.575 [1.824, 3.636] | 1.037 [1.027, 1.044] | 0.575 | <0.001* | 413 | No AD |
| Calcification | 0.744 [0.536, 1.030] | 0.934 [0.849, 1.006] | <0.001* | 0.075 | 1,248 | No calcification |

Univariate two-sample Student's t-test was conducted to compare the difference in the false negative rate of bootstrap performance between subgroups and control groups. Demographic and clinical/imaging features were evaluated using a multivariate logistic regression model for descriptive analysis to control for confounding effects between the selected features. A total of 6,142 patches were inspected. Odds ratios (ORs) and Risk ratios (RRs) are presented with their corresponding 95% confidence intervals (CIs) in brackets.

*Statistically significant, p < 0.05.

AD, architectural distortion; BI-RADS, breast imaging reporting and data system; OR, odds ratio; RR, risk ratio.

facing performance degradation in patch classification may also face degradation in performance in the applications such as tumor object detection software. Lastly, we only included exams in subgroup analysis for which data was available and grouped imaging features and race into broad categories, which may obscure nuanced differences within these groups.

## Conclusion

The study is the first to employ multivariate analysis to explore model failures for a breast cancer patch classification model across multiple subgroups. Our findings advance computational strategies for CADe and CADx applications leveraging deep learning, support the development of more precise and broadly applicable tools, and contribute to fairer and more interpretable decision-making models in mammography while identifying populations that may face disadvantages if no measures are taken. These are important steps towards improving the performance of fair and interpretable decision-making models powered by deep learning in mammography.

**Table 5. Evaluation of model performance using multivariate regression to assess risk ratio of false positive predictions by subgroups, compared to versus univariate evaluation.**

| Variables | OR | RR | Univariate *p*-value | Multivariate *p*-value | Number of patches | Control group |
|---|---|---|---|---|---|---|
| Black | 1.107 [0.911, 1.344] | 1.058 [0.948, 1.17] | 0.153 | 0.306 | 3,071 | White |
| Other | 0.982 [0.751, 1.283] | 0.990 [0.842, 1.143] | <0.001* | 0.893 | 1,045 | White |
| 50–60 y/o | 0.914 [0.725, 1.151] | 0.934 [0.779, 1.11] | <0.001* | 0.445 | 2,028 | <50 y/o |
| 60–70 y/o | 0.870 [0.677, 1.12] | 0.899 [0.736, 1.087] | <0.001* | 0.279 | 1,851 | <50 y/o |
| >70 y/o | 0.920 [0.703, 1.202] | 0.938 [0.759, 1.144] | <0.001* | 0.540 | 1,529 | <50 y/o |
| Density B | 1.328 [0.976, 1.808] | 1.249 [0.981, 1.563] | <0.001* | 0.071 | 2,594 | Density A |
| Density C* | 2.406 [1.799, 3.216] | 1.891 [1.558, 2.251] | <0.001* | <0.001* | 3,043 | Density A |
| Density D* | 3.863 [2.492, 5.989] | 2.486 [1.934, 3.048] | 0.015* | <0.001* | 256 | Density A |

Univariate two-sample Student's t-test was conducted to compare the difference in the false positive rate of bootstrap performance between subgroups and control groups. Demographic and clinical/imaging features were evaluated using a multivariate logistic regression model for descriptive analysis to control for confounding effects between the selected features. Some features were not considered in this model because pathological outcomes and image findings do not exist in false positive cases. A total of 7,248 patches were inspected. Odds ratios (ORs) and Risk ratios (RRs) are presented with their corresponding 95% confidence intervals (CIs) in brackets.

*Statistically significant, p < 0.05.

AD, architectural distortion; BI-RADS, breast imaging reporting and data system; OR, odds ratio; RR, risk ratio.

## Supporting information

**S1 Table. Comparison of performance of multiple standard convolutional neural network (CNN) models for binary patch classification on mammography.**
(DOCX)

**S2 Table. Classification performance in subgroups stratified by tissue density on the test set.**
(DOCX)

## Author contributions

**Conceptualization:** MinJae Woo, Beatrice Brown-Mulry, Imon Banerjee, Laleh Seyyed-Kalantari, Hari Trivedi.

**Data curation:** MinJae Woo, Linglin Zhang, Beatrice Brown-Mulry, InChan Hwang.

**Formal analysis:** MinJae Woo, Linglin Zhang, Beatrice Brown-Mulry, Laleh Seyyed-Kalantari, Hari Trivedi.

**Investigation:** MinJae Woo, Beatrice Brown-Mulry, InChan Hwang, Judy Wawira Gichoya, Aimilia Gastounioti, Imon Banerjee, Laleh Seyyed-Kalantari, Hari Trivedi.

**Methodology:** MinJae Woo, Linglin Zhang, Beatrice Brown-Mulry, Laleh Seyyed-Kalantari, Hari Trivedi.

**Project administration:** MinJae Woo.

**Resources:** Hari Trivedi.

**Supervision:** MinJae Woo, Judy Wawira Gichoya, Aimilia Gastounioti, Imon Banerjee, Laleh Seyyed-Kalantari, Hari Trivedi.

**Validation:** MinJae Woo, Linglin Zhang, Beatrice Brown-Mulry, InChan Hwang, Hari Trivedi.

**Visualization:** MinJae Woo.

**Writing – original draft:** MinJae Woo, Linglin Zhang, Judy Wawira Gichoya, Laleh Seyyed-Kalantari, Hari Trivedi.

**Writing – review & editing:** MinJae Woo, Aimilia Gastounioti, Hari Trivedi.

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
