## [Decision Letter · Decision Letter 0]

14 Jan 2025

PDIG-D-24-00374Subgroup Evaluation to Understand Performance Gaps in Deep Learning-Based Classification of Regions of Interest on MammographyPLOS Digital Health Dear Dr. Woo, Thank you for submitting your manuscript to PLOS Digital Health. After careful consideration, we feel that it has merit but does not fully meet PLOS Digital Health's publication criteria as it currently stands. Therefore, we invite you to submit a revised version of the manuscript that addresses the points raised during the review process. Please submit your revised manuscript within 30 days Feb 13 2025 11:59PM. If you will need more time than this to complete your revisions, please reply to this message or contact the journal office at digitalhealth@plos.org. Please include the following items when submitting your revised manuscript:* A rebuttal letter that responds to each point raised by the editor and reviewer(s). You should upload this letter as a separate file labeled 'Response to Reviewers '. This file does not need to include responses to any formatting updates and technical items listed in the 'Journal Requirements' section below.* A marked-up copy of your manuscript that highlights changes made to the original version. You should upload this as a separate file labeled 'Revised Manuscript with Track Changes '.* An unmarked version of your revised paper without tracked changes. You should upload this as a separate file labeled 'Manuscript '. If you would like to make changes to your financial disclosure, competing interests statement, or data availability statement, please make these updates within the submission form at the time of resubmission. Guidelines for resubmitting your figure files are available below the reviewer comments at the end of this letter. We look forward to receiving your revised manuscript. Kind regards, Chiara Corti, MDAcademic EditorPLOS Digital Health Leo Anthony CeliEditor-in-ChiefPLOS Digital Healthorcid.org/0000-0001-6712-6626  **Journal Requirements:**

1. We have amended your Competing Interest statement to comply with journal style. We kindly ask that you double check the statement and let us know if anything is incorrect. 

 **Additional Editor Comments (if provided):****Reviewers' Comments:** Reviewer's Responses to Questions

**Comments to the Author**

1. Does this manuscript meet PLOS Digital Health’s publication criteria ? Is the manuscript technically sound, and do the data support the conclusions? The manuscript must describe methodologically and ethically rigorous research with conclusions that are appropriately drawn based on the data presented.

Reviewer #1: Yes

Reviewer #2: Yes

2. Has the statistical analysis been performed appropriately and rigorously?

Reviewer #1: Yes

Reviewer #2: I don't know

3. Have the authors made all data underlying the findings in their manuscript fully available (please refer to the Data Availability Statement at the start of the manuscript PDF file)?

Reviewer #1: Yes

Reviewer #2: No

4. Is the manuscript presented in an intelligible fashion and written in standard English?

Reviewer #1: Yes

Reviewer #2: Yes

5. Review Comments to the Author

Reviewer #1: Very good manuscript exploring performance gaps in deep learning models used for classifying normal versus abnormal regions of interest (ROIs) in mammography.

It uses a large and racially representative dataset, allowing for robust model training and enabling thorough subgroup analysis across demographics. This data diversity strengthens the study’s generalizability and the use of multivariate regression models to control for confounding variables enhances the reliability of findings. Moreover, a comprehensive set of evaluation metrics, including AUC, recall, and false positive/negative rates gives a nuanced understanding of the model’s effectiveness. The fact that, however, all data originates from a single institution it correctly mentioned in the limitations of the study.

Major comments:

1. Methods

The manuscript could benefit from additional detail regarding data preprocessing (i.e., handling of demographics imbalances, if present) and description of softwares and tools, or programming languages used in data analysis.

2. Discussion

Including references that discuss practical implications for end-users (e.g., radiologists) on how these model errors impact real-world diagnostic decisions or patient outcomes could strengthen the study’s relevance.

Minor comments:

1. Consider reviewing the following reformulated sentences:

-Breast cancer is the most common cancer among women, leading to approximately 42,000 deaths annually in the United States (Introduction).

- Deep learning (DL) models are increasingly used to improve differentiation between normal and suspected abnormal regions on mammograms (Introduction).

- The dataset provides comprehensive details on demographics, imaging features, pathologic outcomes, and BI-RADS tissue densities for both screening and diagnostic exams (Methods).

- This study contributes to the advancement of fairer and more interpretable decision-making models in mammography (Conclusions).

- Insights from this research can support the development of diagnostic tools that are accurate and equitable (Conclusions).

2. In the caption of Figure 2, add a brief description of the matching process (mentioning the row- and column-wise distribution).

3. In line 9 of the Discussion paragraph, provide also literature examples that support your sentence (i.e., classes are interrelated).

4. Consider specifying that White class is significantly more likely for false negative with respect to other races, but not Black.

Reviewer #2: 1.Citation formatting appears inconsistent in a few places use uniform formatting.

2.Figures are critical to understanding such a technical study and should be improved with high resolution.

3.Confidence intervals for false positive/negative risk rates are not provided. Providing this, if possible, will increase the clarity and reliability of the results.

6. PLOS authors have the option to publish the peer review history of their article (what does this mean? ). If published, this will include your full peer review and any attached files.

**Do you want your identity to be public for this peer review?** For information about this choice, including consent withdrawal, please see our Privacy Policy .

Reviewer #1: No

Reviewer #2: **Yes: ** Elvina Almuradova

---

## [Editor Report · Decision Letter 1]

4 Mar 2025

Subgroup Evaluation to Understand Performance Gaps in Deep Learning-Based Classification of Regions of Interest on Mammography

PDIG-D-24-00374R1

Dear Dr. Woo,

We are pleased to inform you that your manuscript 'Subgroup Evaluation to Understand Performance Gaps in Deep Learning-Based Classification of Regions of Interest on Mammography' has been provisionally accepted for publication in PLOS Digital Health.

Best regards,

Daniel A Hashimoto, MD, MSTR

Section Editor

PLOS Digital Health

**Additional Editor Comments (if provided):**

The authors have appropriately addressed the reviewer comments and request for minor revision. One sentence could benefit from a very minor grammatical correction.

- Author Summary: "Our study reveals that CNN classifier do not fail randomly in mammographic images, resulting in certain patient populations more prone to experiencing these failures." Likely should be "...that CNN classifiers do not fail..."